# Carbodefluorination of fluoroalkyl ketones via a carbene-initiated rearrangement strategy

Linxuan Li [1,5], Xinyu Zhang[1,5], Yongquan Ning[1,5], Xiaolong Zhang[1,5], Binbin Liu[1], Zhansong Zhang[1], Paramasivam Sivaguru [1], Giuseppe Zanoni[2], Shuang Li[1], Edward A. Anderson [3] & Xihe Bi [1,4] ✉

The C–F bond cleavage and C–C bond formation (i.e., carbodefluorination) of readily accessible (per)fluoroalkyl groups constitutes an atom-economical and efficient route to partially fluorinated compounds. However, the selective mono-carbodefluorination of trifluoromethyl ($CF_3$) groups remains a challenge, due to the notorious inertness of C–F bond and the risk of over-defluorination arising from C–F bond strength decrease as the defluorination proceeds. Herein, we report a carbene-initiated rearrangement strategy for the carbodefluorination of fluoroalkyl ketones with β,γ-unsaturated alcohols to provide skeletally and functionally diverse α-mono- and α,α-difluoro-γ,δ-unsaturated ketones. The reaction starts with the formation of silver carbenes from fluoroalkyl *N*-triftosylhydrazones, followed by nucleophilic attack of a β,γ-unsaturated alcohol to form key silver-coordinated oxonium ylide intermediates, which triggers selective C–F bond cleavage by HF elimination and C–C bond formation through Claisen rearrangement of in situ generated difluorovinyl ether. The origin of chemoselectivity and the reaction mechanism are determined by experimental and DFT calculations. Collectively, this strategy by an intramolecular cascade process offers significant advances over existing stepwise strategies in terms of selectivity, efficiency, functional group tolerance, etc.

The construction of C–C bonds is fundamental to the art of organic synthesis as it provides access to the backbone of organic molecules, including pharmaceuticals, agrochemicals, and functional materials[1–5]. The development of new methods, that are advantageous in terms of selectivity, availability and affordability of starting materials, functional group tolerance, and environmental sustainability is a constant focus of organic chemistry. Among numerous C–C bond-forming reactions, dehalogenative cross-coupling reactions have long been regarded as the most reliable and efficient tactics for assembling carbon scaffolds[6–9]. However, in contrast to other C–X bonds (X = Cl, Br, I), the cleavage of a C–F bond and formation of a new C–C bond (so-called carbodefluorination) remain a formidable challenge in modern organic chemistry, especially in readily accessible trifluoromethyl ($CF_3$) groups[10–16]. The main challenge with this respect is the notorious inertness of C–F bonds, accompanied by a decrease in bond dissociation energy (BDE) as defluorination takes place[17], which often results in undesired over-defluorination (Fig. 1a)[10,18–20]. In this context, great efforts have been invested in achieving selective mono-

[1]Department of Chemistry, Northeast Normal University, Changchun 130024, China. [2]Department of Chemistry, University of Pavia, Viale Taramelli 12, 27100 Pavia, Italy. [3]Chemistry Research Laboratory, University of Oxford, 12 Mansfield Road, Oxford OX1 3TA, UK. [4]State Key Laboratory of Elemento-Organic Chemistry, Nankai University, Tianjin 300071, China. [5]These authors contributed equally: Linxuan Li, Xinyu Zhang, Yongquan Ning, Xiaolong Zhang. ✉e-mail: bixh507@nenu.edu.cn

**Fig. 1 | Background and motivation for carbodefluorination of C–F bonds of trifluoromethyl groups. a** Bond Dissociation Energy (BDE) values of four fluorine-substituted methanes. **b** The strategies for selective carbodefluorination of C–F bonds: drawbacks and solutions. **c** This work: a carbene-initiated rearrangement strategy for carbodefluorination of fluoroalkyl ketones. (het)Ar (hetero)aryl, Alk alkyl, Tfs triftosyl.

carbodefluorination of the CF$_3$ group, especially in CF$_3$-substituted arenes and alkenes through the generation of difluoro-substituted carbanion (by low-valent metal or electrochemical reduction)[21–25], carbocation (using strong Lewis acids)[26,27] or radical (using excited state photocatalysts) intermediates by either heterolytic or homolytic cleavage of a C–F bond[28–35], where the in situ generated difluoromethylene reactive intermediates can be stabilized through p–π conjugation (Fig. 1b). While the acyl-CF$_3$ compounds (CF$_3$C=O) are most abundant and easily available, only two classes of selective mono-defluorinative C–C bond-forming reactions of acyl-CF$_3$ derivatives have been disclosed, namely: (i) copper or low-valent magnesium promoted defluorinative coupling of trifluoromethyl ketones with aldehydes or ketones[23,36–38]; and (ii) radical defluorinative coupling of acyl-CF$_3$ with alkenes by spin-center shift (SCS) or photocatalysis[28,34,35], which can be ascribed to the incompatibility between the conditions required for the generation of reactive intermediates and high reactivity of the C=O bond. Nevertheless, these approaches to the carbo-defluorination of CF$_3$-arenes, -alkenes, and -acyl compounds generally require a stepwise mechanism via reactive difluoromethylene inter-mediates, such as carbanions, carbocations, or radicals, which can generally lower the extent of selectivity, efficiency, substrate scope, and functional group tolerance. Through the formation of *N*-sulfonylhydrazones, fluoroalkyl ketones have recently become versatile coupling partners, especially as a source of fluoroalkyl carbene in a range of C–C bond-forming reactions[39–47]. Moreover, the rearrangement reaction of fluorine-containing molecules can provide various fluorine-containing molecular frameworks[48–53]. Hence, the

development of a strategy enabling the integration of successive C–F bond cleavage and C–C bond formation into an intramolecular cascade process would offer significant advantages over existing stepwise strategies.

Here, we report a carbene-initiated rearrangement strategy for the carbodefluorination of fluoroalkyl ketones (Fig. 1c, up). We envisage that an intramolecular rearrangement strategy could provide an advantageous route for the efficient carbodefluorination of trifluoromethyl ketones, i.e., the formation of a metal-coordinated ylide intermediate by nucleophilic attack of β,γ-unsaturated alcohols to a fluoroalkyl metal carbene might enable a sequential C–F bond cleavage / C–C bond formation through the Claisen rearrangement of an in situ generated difluorovinyl ether intermediate (Fig. 1c, bottom). We eventually implement this carbene-initiated rearrangement strategy for selective carbodefluorination of fluoroalkyl ketones through the reaction between their *N*-triftosylhydrazones and β,γ-unsaturated alcohols by silver catalysis. The scope of this transformation includes five-membered (benzo-fused)heteroaryl carbinols, allyl and propargyl alcohols, thus providing access to skeletally and functionally diverse α-mono- and α,α-difluoro-γ,δ-unsaturated (cyclo)alkyl ketones (Fig. 1c, middle).

## Results and discussion
### Substrate scope
Transition-metal-catalyzed dearomative functionalization of (hetero) aromatics has recently emerged as a powerful method to access aliphatic cyclic compounds[54–60]. Dearomative functionalization of

indoles to generate valuable indolines is particularly interesting due to the frequent occurrence of the latter substructures in natural products and other alkaloids[55]. Despite many advances, the formation of new carbon-carbon bonds via defluorinative dearomatization of indoles remains elusive[54,56,57]. The chemical inertness of the C−F bond, and the energetic barrier associated with the disruption of aromaticity are the main factors that prevent the success of the approach via indole defluorinative dearomatization. At the outset, we choose trifluoromethyl ketone-derived *N*-sulfonylhydrazones as diazo surrogates to verify the planned reaction hypothesis, with indole-3-carbinols serving as the nucleophile. A survey of various combinations of different fluoroalkyl *N*-sulfonylhydrazones, metal catalysts, solvents and temperature revealed that a mixture of phenyl trifluoromethyl ketone *N*-triftosylhydrazone (**1a**), indole-3-carbinol (**2aa**), $K_2CO_3$ (2.0 equiv) and $Tp^{Br3}Ag$ (10 mol%) in toluene at 80 °C achieved the desired defluorinative [3,3]-rearrangement product (**3**) in 84% yield (Fig. 2). The driving force to destroy the aromaticity of the substrate comes from the flow of electrons during the opening of the six-membered ring transition state during the [3,3]-rearrangement[61]. With these optimized conditions in hand, we sought to examine the scope of this defluorinative dearomatization reaction with respect to various indole-3-carbinols. As shown in Fig. 2, all the primary indole-3-carbinols that were investigated afforded the desired rearrangement products **3–12** in high yield, regardless of the position and electronic effect of the substituents. A range of secondary indole-3-carbinols was also suitable for this reaction, forming 2-difluoroacylated indolines (**13–18**) in moderate to good yield with excellent stereoselectivity (*E/Z* up to >20:1). However, we observed that tertiary alcohols are not suitable for this transformation, possibly because the structures of tertiary alcohols are more sterically hindered during electrophilic attack. Again, the substitution patterns of the C2-substituted indoles did not affect significantly the reaction outcomes (**19–22**). This strategy thus provides an opportunity for the construction of non-aromatic *N*-heterocycles containing a quaternary carbon center, which is the key precursor for the synthesis of polycyclic complex molecules[59,60,62]. Variation of the *N*-protecting group on the indole-3-carbinol was also well tolerated, including aryl and alkyl *N*-sulfonyl (**23–30**), *N*-Boc (**31**), *N*-Cbz (**32**) and *N*-acetyl (**33**) groups. In addition to indole-3-carbinols, indole-2-carbinols could also be employed without difficulty in this transformation, affording the corresponding products featuring an enamine motif (**34–38**), which are difficult to access with conventional methods[63,64]. Regarding the scope of *N*-triftosylhydrazones, an array of substrates reacted smoothly with indole-3-carbinol, affording the corresponding products (**39–56**) in high yield. Functional groups such as methyl (**39, 40**), *tert*-butyl (**41**), methoxy (**42, 53**), trifluoromethoxy (**43**), phenyl (**44**), halogen (**45–48, 51, 52**), trifluoromethyl (**49**), and vinyl (**50**) were well tolerated. The use of indole-2-carbinol in the reaction with a *p*-Br-phenyl trifluoromethyl ketone *N*-triftosylhydrazone also gave a similar reaction outcome (**57**). It should be noted that *N*-triftosylhydrazones derived from alkyl trifluoromethyl ketone are not suitable substrates, as undergo competitive 1,2-hydrogen shifts to form alkenes.

We were pleased to find that a wide array of other heteroaryl carbinols are also suitable for this dearomatizing rearrangement. In addition to benzofurans and benzothiophenes, a variety of heterocyclic rings with a greater degree of aromaticity such as pyrrole, furan, and thiophene carbinols underwent the defluorinative dearomatization with phenyl trifluoromethyl *N*-triftosylhydrazone to generate the corresponding α,α-difluoro-γ,δ-unsaturated alkyl ketone products (**58–85**) in moderate to excellent yield, with good functional group tolerance. The formation of a quaternary center on a fused-ring furan (**80**) is particularly notable in view of the congested nature of this system. Collectively, these results demonstrate the dearomative carbodefluorination of C(sp3)−F bonds, which constitutes a powerful method for the synthesis of a wide range of α-heterocyclic fluoroalkyl ketones.

Having established a strategy for the highly selective defluorinative functionalization of C−F bonds in the acyl $CF_3$ group with heteroaromatic carbinols, we speculated whether the scope of this transformation could be expanded to other β,γ-unsaturated alcohols. In the event, this chemistry indeed proved suitable for application with these other substrates, with optimization of the carbodefluorination reaction between **1a** and allyl alcohol (**2c**) summarized in Supplementary Tables 2, 3. Under these optimized conditions (Fig. 3), a broad range of 1,1- and 1,2-disubstituted primary allyl alcohols bearing different (cyclo)alkyl and aryl substituents were found to give good to excellent yield of the desired products (**86–100**). Functionalities such as halogens (**91, 92, 99**), alkene (**93**) and alkyne (**94**) motifs remain intact, allowing for further derivatizations. Trisubstituted allyl alcohols possessing styryl (**101**), alkenyl chloride (**102**), alkenyl ester (**103**), and (hetero)cyclic alkene (**104, 105**) also performed well. Similar to primary allyl alcohols, secondary alcohols bearing functionalities such as cycloalkyl (**106, 115**), phenyl (**107**), ester (**108**), piperidine (**109**), alkenyl (**110, 113**), allyl ether (**111**), ketal (**112**) and methyl (**114, 117**) groups at the α-position were suitable for this transformation, affording a range of disubstituted homoallylic α,α-difluoroketones (**106–117**) in 53–94% yield. α,α-Difluoro-γ,δ-enones with a fluoroalkyl all-carbon quaternary center could be prepared (**118–122**) in moderate to excellent yield using 1,1-disubstituted allyl alcohols; such α-difluoroalkyl quaternary carbons, which cannot be easily prepared by conventional methods[65,66]. This protocol was also applicable for the late-stage modification of natural products, such as myrtenol (**123**), geraniol (**124**) and insect repellent cyclocitral (**125**) with high selectivity and high yield.

The propargylic Claisen rearrangement is a powerful method for the synthesis of allenes[67–69]. With this in mind, we next examined the carbodefluorination of trifluoromethyl ketones with propargyl alcohols, with a view to accessing difluoroalkyl-substituted allenes, which are gaining importance in drug discovery. We were delighted to find that a variety of propargyl alcohols provided the desired α-$CF_2$ allenyl products in good yield and selectivity (**126–154**). To our knowledge, this is the first example of selective C−F allenylation of $CF_3$ groups[10–16]. This chemistry proved most effective using propargyl alcohol itself, which exhibited high reactivity and gave the desired product (**126**) in near quantitative yield. However, also both acylic (linear or branched) and cycloalkyl-substituted propargyl alcohols smoothly afforded the corresponding allenes in 78–97% yield (**127–130**). Pleasingly, propargyl alcohols bearing phenyl (**131**), naphthyl (**132**), thienyl (**133**), TMS (**134**) and halogen (**135, 136**) groups on the alkyne terminus proved suitable substrates, providing the corresponding products in good yield in almost all cases. TMS, Cl or Br functionalities could be retained in the products, which allows for further orthogonal functionalization of the thus obtained products. Alkyl propargyl alcohols containing various functionalities on the alkyl sidechain, such as phenyl (**137**), chloro (**138**), ether (**139**), ester (**140**), pyran (**141**), alkenyl (**142**), and alkynyl (**143**) groups, were all amenable in this reaction. Notably, the free alkenyl and alkynyl units in **142** and **143** were untouched, which demonstrates the high selectivity of the alcohol hydroxyl (-OH) group toward metal carbene trapping to form the proposed intermediate silver-coordinated oxonium ylides[42]. Polyfunctionalized allenes were found to be hard to prepare by existing methods[69–73]. Furthermore, a cyclohexene-conjugated difluoroalkyl allene (**144**), a useful synthon in cycloaddition reaction to access polycyclic fluorinated molecules, was also obtained in 80% yield from the corresponding enyne. Secondary propargyl alcohols proved to be similarly suitable substrates, providing diverse disubstituted and trisubstituted allenes (**145–152**). Notably, this strategy enables access to structures bearing pharmaceutically relevant sidechains, such as floramelon (**153**) and citronellal (**154**).

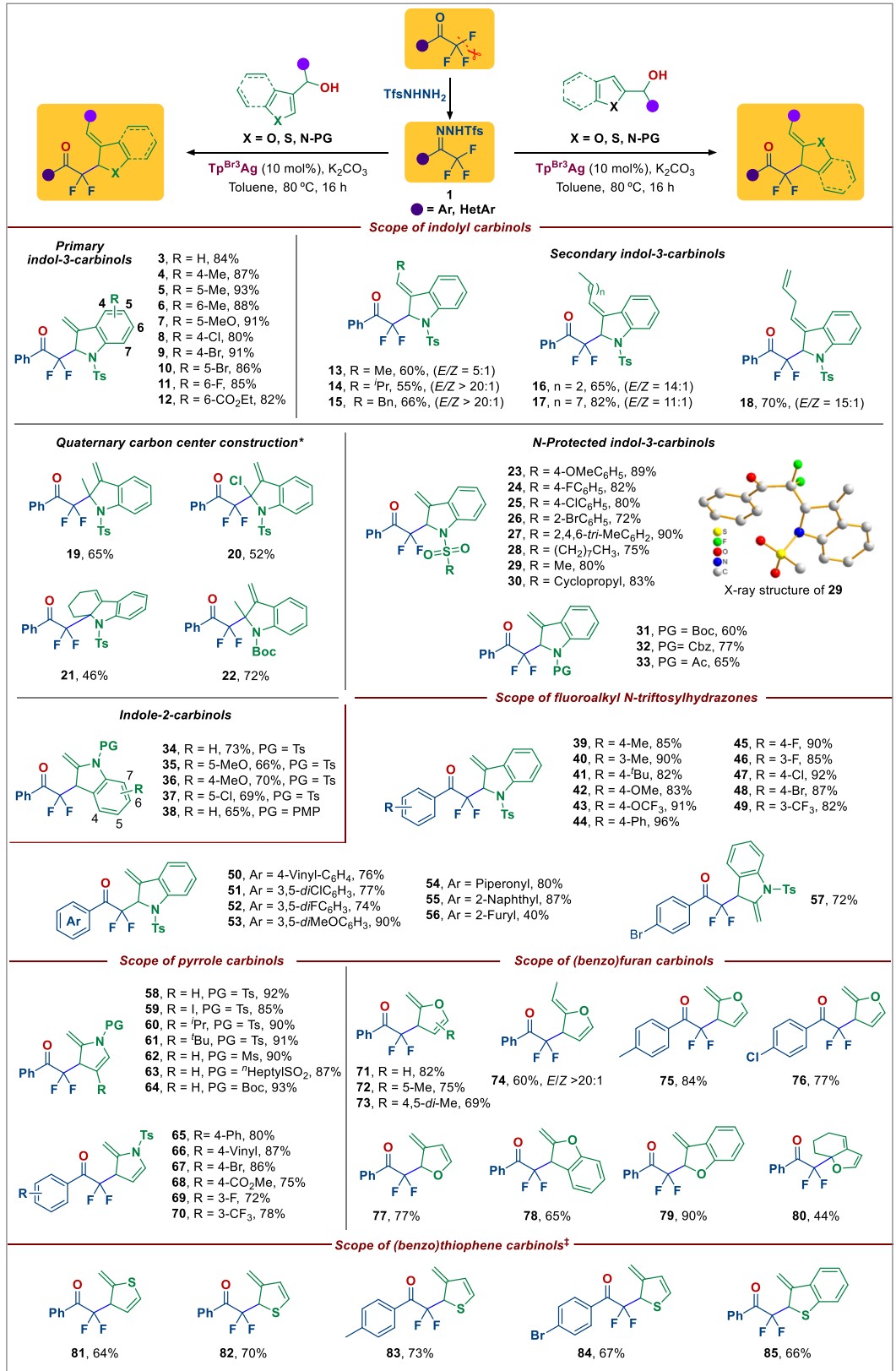

**Fig. 2 | Scope of defluorinative dearomatization of indole carbinols with fluoroalkyl *N*-triftosylhydrazones.** Reaction conditions: all reactions were carried out with **1** (0.3 mmol, 1.0 equiv), **2a** or **2b** (0.6 mmol, 2.0 equiv), K₂CO₃ (0.6 mmol, 2 equiv) and TpBr3Ag (10 mol%) in toluene (4 mL) at 80 °C for 16 h. *Reaction were carried out at 80 °C for 8 h, then 120 °C for 24 h. †DCM was used instead of toluene. ‡Cs₂CO₃ was used instead of K₂CO₃ under 100 °C. All yield refers to isolated yield. PG protecting group, Ts tosly, Boc t-butoxycarbonyl, Cbz carbobenzyloxy, Ac acetyl, PMP p-methoxyphenyl, Ms methanesulfonyl.

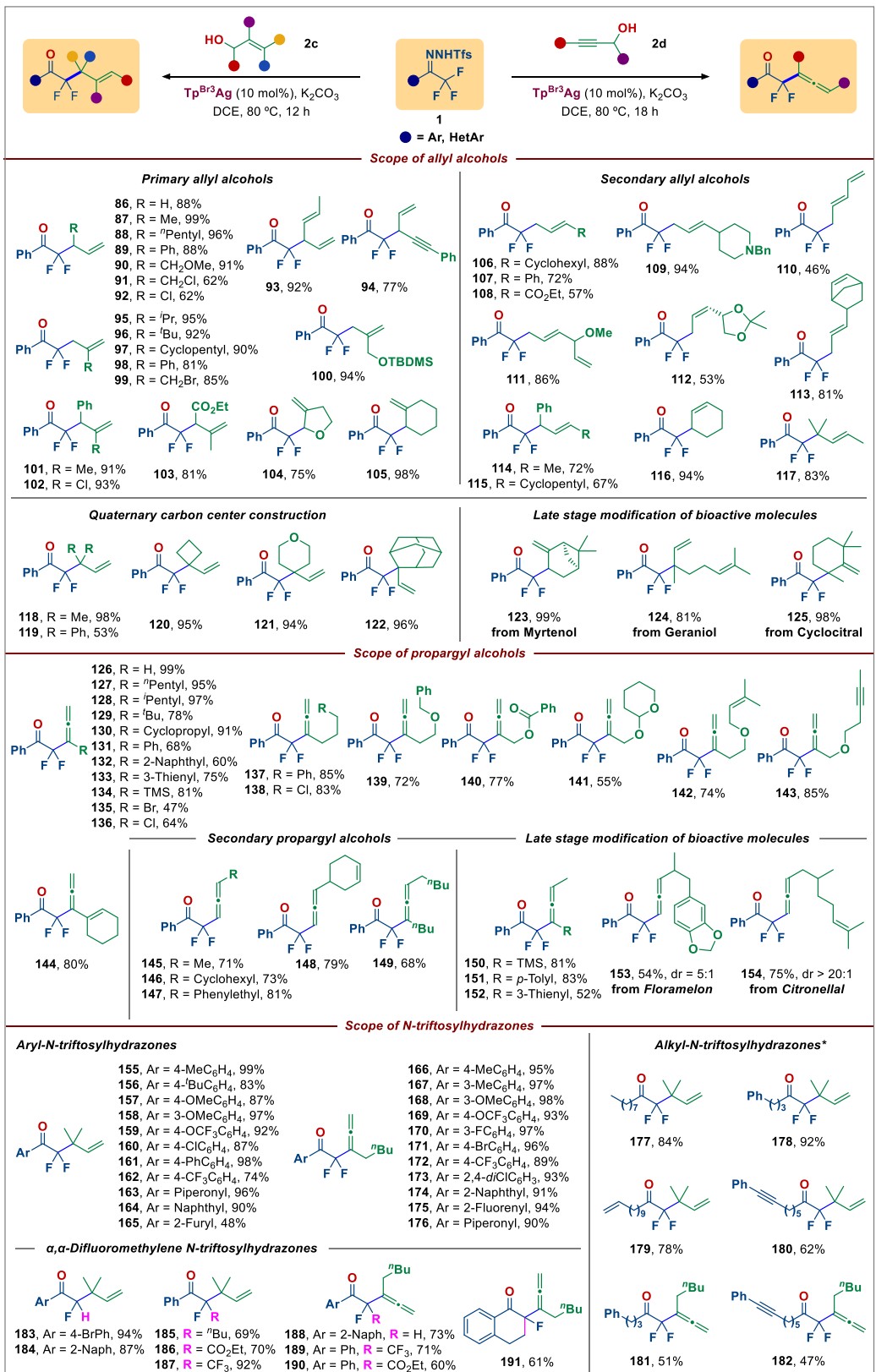

**Fig. 3 | Scope of Defluorinative Allylation.** Reaction conditions: all reactions were carried out with **1** (0.3 mmol, 1.0 equiv), **2c/2d** (0.6 mmol, 2.0 equiv), $K_2CO_3$ (0.6 mmol, 2 equiv) and $Tp^{Br3}Ag$ (10 mol%) in 1,2-dichloroethane (DCE) (4 mL) at 80 °C. Isolated yields. ***1** (0.3 mmol, 1.0 equiv), **2** (0.6 mmol, 2.0 equiv), $N,N$-diisopropylethylamine (DIPEA) (0.6 mmol, 2 equiv) and $Rh_2(esp)_2$ (2 mol%) in DCE (4 mL) at 80 °C. Ar aryl, HetAr heteroaryl, TBDMS tert-butyldimethylsilyl, Bn benzyl, TMS trimethylsilyl, nBu n-butyl, dr diastereomeric ratio.

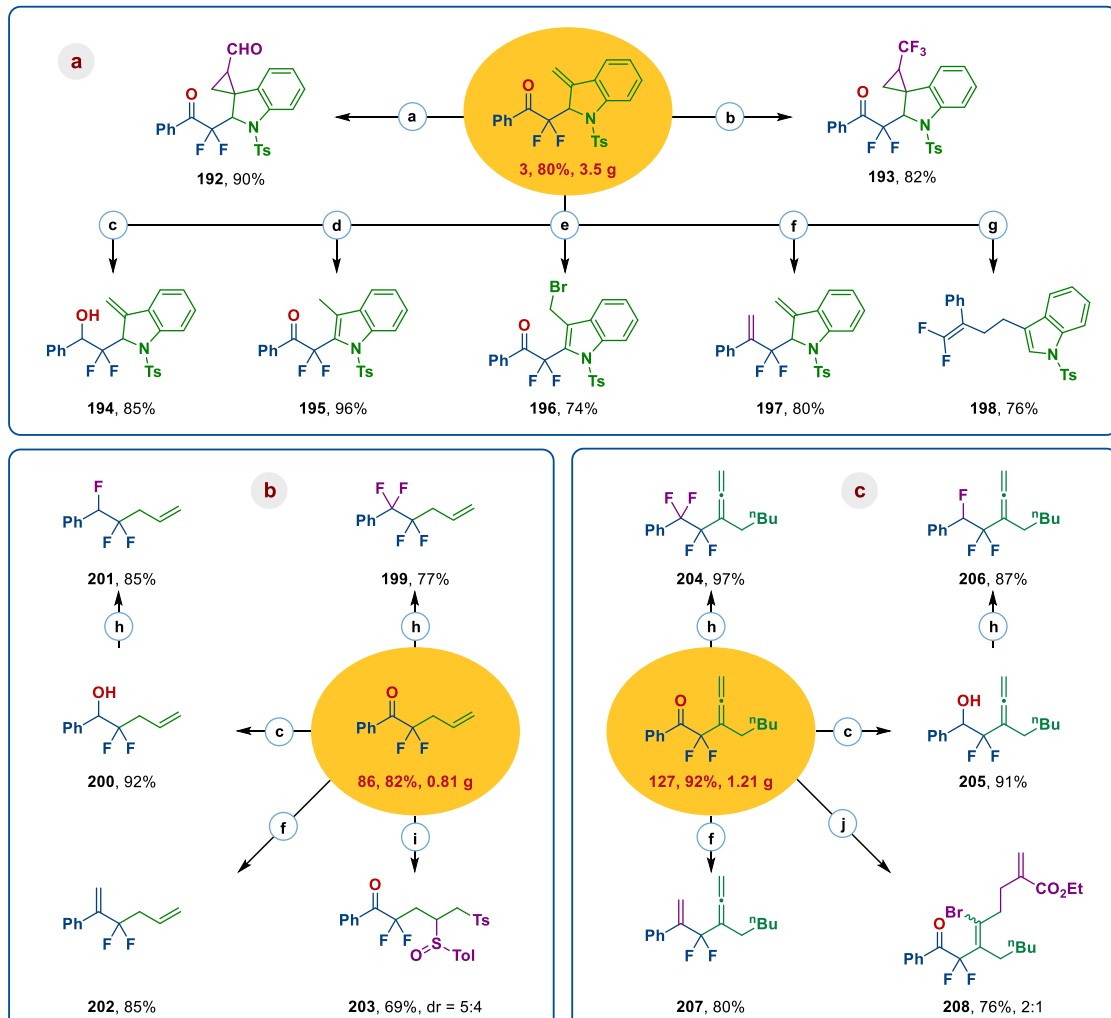

**Fig. 4 | Gram-scale synthesis and further transformation. a** Gram-scale synthesis and further transformation of compound **3**; **b** Gram-scale synthesis and further transformation of compound **86**; **c** Gram-scale synthesis and further transformation of compound **127**. Reaction conditions: a. **3** (0.2 mmol), difluoroacetaldehyde *N*-triftosylhydrazone (DFHZ-Tfs) (0.4 mmol), FeTPPCl (3 mol%), aqueous NaOH (5.0 wt%)/toluene, 60 °C; b. **3** (0.2 mmol), trifluoroacetaldehyde *N*-tfsylhydrazone (TFHZ-Tfs) (0.4 mmol), FeTPPCl (3 mol%), K$_2$CO$_3$ (0.6 mmol), 1,4-dioxane (3 mL), 40 °C; c. **3/86/127** (0.2 mmol), NaBH$_4$ (0.24 mmol) in CH$_3$OH (2 mL) at 25 °C; d. NaI (0.32 mmol), TMSCl (0.32 mmol), H$_2$O (0.16 mmol), CH$_3$CN (2 mL), room temperature; e. **3** (0.3 mmol), NBS (1.1 equiv), DCM (2 mL), 0 °C - rt; f. **3/86/127** (0.2 mmol), Ph$_3$PCH$_3$Br (0.4 mmol), $^t$BuOK (0.4 mmol), THF (2 mL), 25 °C; g. **3** (0.2 mmol), Ph$_3$PCH$_3$Br (0.3 mmol), $^t$BuOK (0.3 mmol), THF (2 mL), 25 °C - 60 °C; h. **86/127/200/205** (0.2 mmol), DAST (0.2 mmol) in DCM (2 mL) at −78 °C - rt; i. **86** (0.2 mmol), TolSO$_2$Na (1.8 mmol), CH$_3$COCl (1.2 mmol), CHCl$_3$ (2 mL), 10 °C; j. **127** (0.2 mmol), AIBN (0.04 mmol), ethyl bromomethacrylate (0.4 mmol), toluene (2 mL), 80 °C. Ts tosyl, $^n$Bu n-butyl, Tol tolyl, dr diastereomeric ratio.

Finally, we turned our attention to fluoroalkyl ketone *N*-triftosylhydrazones. Both electron-withdrawing and -donating groups on the phenyl ring of the *N*-triftosylhydrazones showed little influence on the reaction outcome and in all instances the desired C−F allylated (**155**–**162**) and C−F allenylated (**166**–**173**) products were obtained in good to excellent yield. Piperonyl (**163, 176**), naphthyl (**164, 174**), furyl (**165**), and fluorenyl (**175**) *N*-triftosylhydrazones were also found to be suitable starting materials. We note that superior efficiency was observed using (Rh$_2$(esp)$_2$ as catalyst instead of Tp$^{Br3}$Ag in the reaction of *N*-triftosylhydrazones derived from alkyl trifluoromethyl ketones (**177**–**182**). Remarkably, the reaction was not restricted to α-trifluoromethyl *N*-triftosylhydrazones. In fact, hydrazones derived from α,α-difluoromethyl (**183, 184, 188**) and α,α-difluoropentyl (**185**) ketones, a α,α-difluoroketoester (**186, 190**) and a difluorocycloalkyl (**191**) ketone were also capable of undergoing this coupling/rearrangement reaction, providing unprecedented opportunities to access a broad array of chemical diversity under a single reactivity platform. In the case of pentafluoroethyl ketone-derived *N*-triftosylhydrazones, the α-C−F bond could be converted to the corresponding allylated (**187**) and allenylated products (**189**) in 92% and 71% yield, respectively, featuring α-fluoro-β-trifluoromethyl functionality.

## Gram-scale reaction and further transformations

The above results demonstrate that readily available α-fluoroalkyl ketones can be converted into a wide variety of valuable α,α-difluoro-γ,δ-unsaturated (cyclo)alkyl ketones with diverse substitution patterns through a silver carbene-initiated defluorination and rearrangement cascade of the corresponding sulfonyl hydrazones. Most of these compounds are newly synthesized and inaccessible by other conventional methods[74–76]. To test the scalability and practicality of this protocol, the gram-scale synthesis of **3**, **86**, and **127** were carried out with the standard set of conditions that we have developed, providing the corresponding products with synthetic efficiency equivalent to the smaller-scale reactions (Fig. 4). Given the importance of α,α-difluoroketones as privileged substructures in medicinal chemistry and the versatile reactivity of carbonyl, heterocyclic, vinyl, and

allenyl moieties, these products could be easily transformed into a broad range of fluorinated building blocks of medicinal relevance. For example, the terminal alkene unit of dearomatization product **3** could be readily cyclopropanated with formyl or trifluoromethyl diazomethanes, affording the corresponding spiroindolines (**192**, **193**) in excellent yield. Furthermore, carbonyl reduction, alkene hydrogenation, alkene bromination, and carbonyl olefination of **3** were achieved with good efficiency (**194–197**), while combining olefination with aromatizing (3,3)-sigmatropic rearrangement offers an attractive entry to 1,1-difluoroalkene products (**198**). The selective nucleophilic *gem*-difluorination of the carbonyl group of **86** and **127** with DAST provided the corresponding tetrafluoro products (**199**, **204**) in 77% and 97% yield, respectively. Compounds **86** and **127** were readily reduced to alcohols in the presence of $NaBH_4$, which enabled monofluorination of products **200** and **205** with DAST to afford the trifluoroalkylated products **201** and **206** in high yield. These conversions enable the synthesis of products with tuneable multivicinal fluorination[77–79]. This platform is attractive for the site-specific introduction of fluorine in aliphatic chains. Notably, the secondary fluoroalkyl alcohol units in compounds **194**, **200**, and **215** are important motifs in bioactive molecules[80]. Carbonyl alkenylation of **86** and **127** gave the desired products **202** and **207** in 85% and 80% yield, respectively. Finally, the radical difunctionalization of olefins (**86**) and allenes (**127**) reliably provided products **203** (69% yield, dr = 5:4) and **208** (76%, stereoselectivity 2:1), respectively.

## Mechanistic investigations

Mechanistic experiments and computational studies were conducted to explore the mechanism of this cascade carbodefluorination process. The progress of the reaction depicted in Fig. 5a was first examined by [1]H NMR. This showed initial formation of intermediate **209**, which reached maximum intensity within an hour. This was transformed to give product **3**, the latter being the near sole reaction component by 16 h. This result suggests that rapid *gem*-difluoroalkenylation is a critical step for the success of this reaction, with this reactive intermediate **209** readily undergoing Claisen rearrangement to afford the final product. Indeed, the subjection of isolated **209** (62% yield (after 40 min), Fig. 5b, eq. 1) to the reaction conditions resulted in 91% yield of product **3**, while the reaction of this intermediate in the absence of silver catalyst afforded **3** in 64% yield (Fig. 5b, eq. 2). These results suggest that the Ag catalyst plays a critical role in formation of the difluoroalkene intermediate, and also facilitates the rearrangement process. A control experiment showed that exposing pre-prepared ether **210** to the standard conditions failed to give the defluorinative rearrangement product **44** (Fig. 5b, eq. 3). This result excluded the possibility of forming an ether intermediate through O–H carbene insertion[81]. Overall, these results suggest that HF elimination to form a *gem*-difluorinated vinyl ether is more favorable than the 1,2-H transfer process of metal ylide to give an ether.

Density functional theory (DFT) calculations at the SMD(toluene)//B3LYP-D3/def2tzvp level of theory were carried out to rationalize the proposed pathway, with the reaction between **1a** and indole-3-carbinol **2aa** selected as a model. As summarized in Fig. 5c, this pathway involves nucleophilic attack, C–F bond cleavage, and [3,3] rearrangement. Compound **1a** is known to undergoes easily a base-mediated decomposition to form a diazo species, which then reacts with $Tp^{Br3}Ag$ catalyst to give a silver carbene[44]. The energy barrier for generation of a silver-coordinated oxonium ylide **Int2** by reaction of indole-3-carbinol with this silver carbene is low (2.6 kcal mol$^{-1}$). The NPA charge analysis of **Int2** shows the F atoms carry more negative charge than the carbene carbon atom, which facilitates abstraction of the hydroxyl proton to form HF. This occurs via a 5-membered ring transition state **TS2** to generate the *gem*-difluorovinyl ether intermediate **Int4** by single bond rotation of initially formed silver-associated *gem*-difluorovinyl ether intermediate **Int3**. Notably, the

energy barrier for the HF elimination is lower ($\Delta\Delta G^\ddagger = 11.8$ kcal mol$^{-1}$) than proton transfer to form an O–H insertion product ($\Delta\Delta G^\ddagger = 20.8$ kcal mol$^{-1}$) (see Supplementary Fig. 7 for details), which is in good agreement with the experimental observations above. Eventually, formation of product **3** takes place by silver-promoted [3,3] rearrangement from **Int4** via **TS3**, which possesses an energy barrier of $\Delta G^\ddagger = 15.5$ kcal mol$^{-1}$ and constitutes the rate-determining step. However, in the absence of silver catalysis, the energy barrier for this step is as high as 20.7 kcal mol$^{-1}$. To explain this reactivity difference, the NPA charge analysis of **Int4** and **Int4'** were carried out. We found that the O–Ag weak coordination in **Int4**, which is absent in **Int4'**, enhances the $C_1$–O bond polarity (NPA charge differences: 0.46 in **Int4** *vs* 0.39 in **Int4'**), thus weakens this bond in **Int4** and makes it easier to break (1.48 Å *vs* 1.45 Å). Furthermore, the color-filled reduced density gradient (RDG) isosurface analysis[82,83] indicate the presence of a strong stabilizing interaction between Ag and O atoms, and also a weak Br···π interaction between the ligand and the benzene ring, both can stabilize the transition state **TS3** (Fig. 5C) (for **TS3'** RDG isosurface, see Supplementary Fig. 8). In a word, all of these factors facilitate the silver-catalyzed [3,3] rearrangement.

In summary, we have established a carbene-initiated rearrangement strategy for the carbodefluorination of fluoroalkyl ketones by the merger of silver catalysis and fluoroalkyl *N*-triftosylhydrazones. This method enables the integration of successive C–F bond cleavage and C–C bond formation on a single molecule entity through a silver carbene-triggered defluorination and rearrangement cascade, including sequential carbene generation, nucleophilic attack, C–F bond cleavage, and eventual C–C bond formation through Claisen rearrangement of resultant difluorovinyl ethers. A broad range of (hetero)aryl/alkyl fluoroalkyl ketones and β,γ-unsaturated alcohols (heteroaryl (indole, pyrrole, (benzo)furan, (benzo)thiophene) carbinols, allyl alcohols and propargyl alcohols) were all found to be amenable to this silver-catalyzed protocol, thereby allowing single-step access to skeletally and functionally diverse α-mono- and α,α-difluoro-γ,δ-unsaturated ketones. These highly functionalized fluorinated molecules will be of great interest as building blocks in drug discovery and materials science. Overall, we believe that this work has opened an avenue to exploit the carbodefluorination of C(sp$^3$)–F bonds.

## Methods

### General procedure for the synthesis of $Tp^{Br3}Ag$[44]

Pre-sublimated 1*H*–3,4,5-tribromopyrazole (40 mmol, 4.0 equiv) and $TlBH_4$ (10 mmol, 1.0 equiv) were added to a 250 mL Schlenk tube and mix well (no magnetic stir required). The tube was fitted with a reflux condenser and a nitrogen balloon (to balance the increased pressure of hydrogen production during the reaction), and three vacuum/nitrogen cycles were made. The reaction was heated at 180 °C for 2 h, then the temperature was raised to 200 °C and the reaction was continued for 2 h. After cooling to room temperature, unreacted pyrazole was removed by vacuum sublimation (150 °C, 2 mbar) to give $TlTp^{Br3}$ as a white solid. AgOTf (10 mmol, 1.0 equiv) and $TlTp^{Br3}$ (10 mmol, 1.0 equiv) were added to the acetone solution to dissolve. After stirring in the dark for 20 h, a white solid precipitated from the initially colorless solution. The solid was filtered off and dried under vacuum for 12 h to give the complex $[Tp^{Br3}Ag]_2 \cdot CH_3COCH_3$. $[Tp^{Br3}Ag]_2 \cdot CH_3COCH_3$ (10 mmol) was stirred in freshly distilled tetrahydrofuran (100 mL) for 30 min in the dark. The solvent was removed under reduced pressure, and white solid $Tp^{Br3}Ag$ was quantitatively obtained after vacuum drying.

### General procedure carbodefluorination reaction of indole-3-carbinols

To a dried sealed tube was charged with *N*-tfsylhydrazone (0.3 mmol, 1.0 equiv), indole-3-carbinol (0.6 mmol, 2.0 equiv), $Tp^{Br3}Ag$ (10 mol%), $K_2CO_3$ (0.6 mmol, 2.0 equiv) in an argon-filled glovebox. Anhydrous

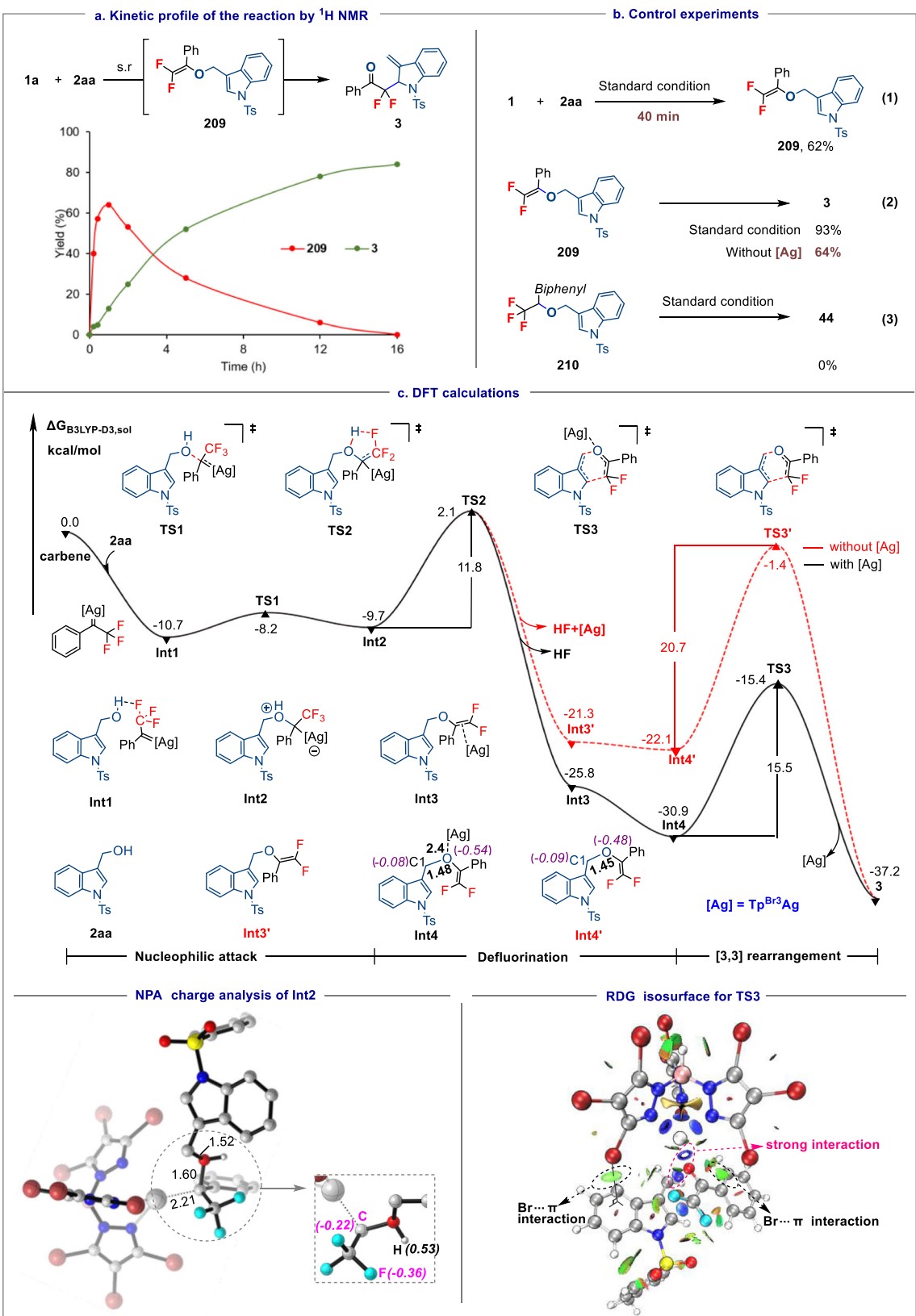

**Fig. 5 | Mechanistic experiments and computational studies. a** Reaction kinetics study; **b** Control experiments; **c** A plausible mechanism based on DFT-computed free-energy profile (ΔG, in kcal·mol⁻¹). Standard condition: **1a** (0.3 mmol, 1.0 equiv),

**2aa** (0.6 mmol, 2.0 equiv), K₂CO₃ (0.6 mmol, 2 equiv) and TpBr3Ag (10 mol%) in toluene (4 mL) at 80 °C. Ts tosyl.

toluene (4 mL) was added. The resulting mixture was stirred at 80 °C for 16 h. When the reaction was completed, the crude reaction mixture was allowed to reach room temperature, and filtered through a short pad of diatomite with ethyl acetate (EtOAc) as an eluent. The filtrate was concentrated in vacuo, and the resulting crude product was purified by column chromatography using ethyl acetate/petroleum ether (1: 25; v: v) to obtain the product.

### General procedure carbodefluorination reaction of 2-substituted indole-3-carbinols

To a dried sealed tube was charged with *N*-tfsylhydrazone (0.3 mmol, 1.0 equiv), 2-substituted indole-3-carbinols (0.6 mmol, 2.0 equiv), Tp$^{Br3}$Ag (10 mol%), K$_2$CO$_3$ (0.6 mmol, 2.0 equiv) in an argon-filled glovebox. Anhydrous toluene (4 mL) was added. The resulting mixture was stirred at 80 °C for 8 h, then the temperature was increased to 120 °C and stirring was continued for 24 h. When the reaction was completed, the crude reaction mixture was allowed to reach room temperature, and filtered through a short pad of diatomite with ethyl acetate (EtOAc) as an eluent. The filtrate was concentrated in vacuo, and the resulting crude product was purified by column chromatography using ethyl acetate/petroleum ether (1: 25; v: v) to obtain the product.

### General procedure carbodefluorination reaction of pyrrole and (benzo)furan carbinols

To a dried sealed tube was charged with *N*-tfsylhydrazone (0.3 mmol, 1.0 equiv), Tp$^{Br3}$Ag (10 mol%), K$_2$CO$_3$ (0.6 mmol, 2.0 equiv) in an argon-filled glovebox. Anhydrous dichloromethane (DCM) (3 mL) was added, then added pyrrole or (benzo)furan carbinols (0.9 mmol, 3.0 equiv) dissolved in DCM (1 mL). The resulting mixture was stirred at 80 °C for 12 h. When the reaction was completed, the crude reaction mixture was allowed to reach room temperature, and filtered through a short pad of diatomite with EtOAc as an eluent. The filtrate was concentrated in vacuo and then the resulting crude product was purified by column chromatography using petroleum ether as eluent to obtain the product. It needs to be treated in a neutral or alkaline medium, and it is easy to restore the aroma under acidic conditions.

### General procedure carbodefluorination reaction of (benzo)thiophene carbinols

To a dried sealed tube was charged with *N*-tfsylhydrazone (0.3 mmol, 1.0 equiv), Tp$^{Br3}$Ag (10 mol%), Cs$_2$CO$_3$ (0.6 mmol, 2.0 equiv) in an argon-filled glovebox. Anhydrous toluene (3 mL) was added, then added (benzo)thiophene carbinols (0.9 mmol, 3.0 equiv) dissolved in toluene (1 mL). The resulting mixture was stirred at 100 °C for 16 h. When the reaction was completed, the crude reaction mixture was allowed to reach room temperature, and filtered through a short pad of diatomite with EtOAc as an eluent. The filtrate was concentrated in vacuo and then the resulting crude product was purified by column chromatography using petroleum ether as eluent to obtain the product. It needs to be treated in a neutral or alkaline medium, and it is easy to restore the aroma under acidic conditions.

### General procedure carbodefluorination reaction of β,γ-unsaturated alcohols

To a dried sealed tube was charged with *N*-tfsylhydrazone (0.3 mmol, 1.0 equiv), Tp$^{Br3}$Ag (10 mol%), K$_2$CO$_3$ (0.6 mmol, 2.0 equiv) in an argon-filled glovebox. Anhydrous 1,2-dichloroethane (DCE) (3 mL) was added, then added allyl alcohols (or propargyl alcohols) (0.6 mmol, 2.0 equiv) dissolved in DCE (1 mL). The resulting mixture was stirred at 80 °C for 12 h. When the reaction was completed, the crude reaction mixture was allowed to reach room temperature, and filtered through a short pad of diatomite with EtOAc as an eluent. The filtrate was concentrated in vacuo and then the resulting crude product was purified by column chromatography using petroleum ether as eluent to obtain the product. (The reaction time for propargyl alcohols is 18 h.)

### General procedure carbodefluorination reaction of alkyl-N-triftosylhydrazones with β,γ-unsaturated alcohols

To a dried sealed tube was charged with alkyl-*N*-tfsylhydrazone (0.3 mmol, 1.0 equiv), Rh$_2$(esp)$_2$ (2 mol%) in an argon-filled glovebox. Anhydrous 1,2-dichloroethane (DCE) (3 mL) was added, then added DIPEA (0.6 mmol, 2.0 equiv) and allyl alcohols (or propargyl alcohols) (0.6 mmol, 2.0 equiv) dissolved in DCE (1 mL). The resulting mixture was stirred at 80 °C for 12 h. When the reaction was completed, the crude reaction mixture was allowed to reach room temperature, and filtered through a short pad of diatomite with EtOAc as an eluent. The filtrate was concentrated in vacuo and then the resulting crude product was purified by column chromatography using petroleum ether as eluent to obtain the product.

## Data availability

The data that support the findings of this study are available within the paper and its Supplementary Information and Supplementary Data files. Raw data are available from the corresponding author on request. Materials and methods, computational studies, experimental procedures, characterization data, $^1$H, $^{13}$C, $^{19}$F NMR spectra, and mass spectrometry data are available in the Supplementary Information. Supplementary Data File 1 contains the cartesian coordinates and energies for the computed structures.

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

## Acknowledgements

This work was supported by NSFC (21871043, 21961130376, 22101044), Postdoctoral Innovation Talent Support Program (BX20200079), Department of Science and Technology of Jilin Province (20190701012GH, 20200801065GH), Fundamental Research Funds for the Central Universities (2412020FZ006). X.B. and E.A. thank the Newton Trust for support (NAF\R1\191210).

## Author contributions

L.L., X-Y.Z., Y.N. and X-L.Z. contributed equally to this work. L.L., X-Y.Z., Y.N., X-L.Z., B.L., Z.Z., P.S. and S.L. performed the experimental investigations and theoretical calculations. L.L., X-Y.Z., Y.N., X-L.Z. and X.B. conceived the concept, designed the project, analyzed the data, and together with P.S., G.Z. and E.A. discussed the results and prepared this manuscript.

## Competing interests

The authors declare no competing interests.
