## [Peer Review File · Nature Communications]

Carbodefluorination of fluoroalkyl ketones via a carbene-initiated rearrangement strategyREVIEWER COMMENTS

Reviewer #1 (Remarks to the Author):

Bi and co-workers reported a formal selective C-F bond allylation/allenylation of trifluoromethylhydrazones by Ag-catalysis to access a diverse range of α,α -difluoroalkylketones featuring heterocycles or alkene/allene moiety. A cascading process including Ag-mediated defluorinative oxygenation and fluorine-Claisen rearrangement was supposed as the success of the reaction, which is distinct from the well-known single electron manifold employed for analogous transformation of trifluoromethyl carbonyls. Significantly broad substrate scope was demonstrated with respect to both of the coupling partners. Also, the gram scale reaction and elaboration of the products highlight the synthetic potency of this reaction. Interestingly, control experiments and DFT calculation reveal the dual role of Ag catalyst to enable the formation of gem-difluoroalkene intermediate and facilitate the subsequent Claisen rearrangement. Overall, this review supports the publication of manuscript in Nat. Commun., after some revisions were addressed.

(1) Does fluorine atoms exert a positive effect on the [3,3]-rearrangement of gem-difluorovinyl ether? Could the reaction proceed under the same conditions when replacing fluoride with chloride or bromide?

(2) Some related work and reviews concerning rearrangements of fluorinated compounds should be cited, for example, Tetrahedron Lett. 1985, 26, 2861; J. Fluorine Chem. 1999, 100, 147; J. Fluorine Chem. 2004, 125, 1593; Tetrahedron 2009, 65, 9905; Org. Biomol. Chem. 2015, 13, 7351, etc. Besides, a latest carbodefluorination reaction reported by the Feng group (Angew. Chem. Int. Ed. 2021, 60, 20237), which also proceeded via a [3,3]-rearrangement, is suggested to be cited.

(3) In Figure 2, only indole, benzofurans, benzothiophenes, pyrrole, furan and thiophene derived carbinols were presented, what about substrates with greater degree of aromaticity, such as benzyl alcohol or naphthalene carbinol?

(4) The authors are suggested to comment on the driving force for the [3,3]-rearrangement which destroy the aromaticity of the substrate. Why the dearomatized products did not undergo rearrangement spontaneously to regain aromaticity?

(5) All presented alcohol substrates were primary and secondary alcohols, are tertiary alcohols viable substrates?

(6) Aside from alcohols, are other nucleophiles, such as amine and thiol suitable for this carbene insertion/rearrangement cascade?

(7) In the DFT part, the calculation result showed that the weak coordination of silver and oxygen could lower the energy barrier for the rate-determining rearrangement step. Could other stronger Lewis acids be applied instead of silver to further reduce the rearrangement energy barrier, resulting in even milder reaction conditions for the [3,3]-rearrangement of gem-difluorovinyl ether?

(8) While N-triflylhydrazone substrates derived from alkyl trifluoromethyl ketones reacted with β,γ -unsaturated alcohols readily, no such a case was demonstrated in Fig. 2 when reacted with heteroaryl carbinols, why?

(8) In cases 177-182, why $Rh_2(esp)_2$ exhibited higher efficiency than $TpBr_3 Ag$? Does any alkene product derived from deprotonation of silver carbene was observed?

(9) In Feng's work (ACIE 2021, 60, 20237), the [3,3]-rearrangement tend to yield CF₂-bridged 1,5-diene product. However, in this work, transformation of 3 to 198, gem-difluoroalkene was obtained as the rearranged product, Please provide an explanation for this.

(10) Last sentence of the abstract, "...in term of..." should be "... in terms of...".

Reviewer #2 (Remarks to the Author):

Very recently the synthesis of fluorinated organic compounds using the acyl-CF₃ compounds (CF₃C(O)R) as fluorine source through carbodefluorination strategy has received a great attention and big progress has been made. But I think this strategy is meaningless because acyl-CF₃ compounds are not easily prepared. That is: the introduction of fluorine atom into organic compounds was a challenging and hard, although the chemistry of the selective defluorination is interesting, the synthesis of new fluorinated compound through defluorination of fluorinated compounds was the

waste of fluorine source. In this manuscript, Bi and co-workers described the carbodefluorination of acyl-CF₃ compounds via fluoroalkyl N-triflylhydrazones. This reaction proceeded through the silver-catalyzed sequential carbene generation, oxonium ylide formation, C–F bond cleavage, and eventual C–C bond formation through Claisen rearrangement of the resultant difluorovinyl ethers. A broad range of (hetero)aryl/alkyl fluoroalkyl ketones and β,γ -unsaturated alcohols were amenable to this reaction. This silver-catalyzed protocol provided single-step access to skeletally and functionally diverse α -mono- and α,α -difluoro- γ,δ -unsaturated ketones. The chemistry described in this manuscript is novel and interesting. Furthermore this manuscript was organized and written well. As I mentioned above, this method has less potential application in medicinal chemistry. But in light of the novelty of chemistry, I recommended for publication of this manuscript in Nature Communications when the following comments were addressed.

- 1) The title of this manuscript was too general. What kind of compound for carbodefluorination should be given.
- 2) How about the reaction of N-triflylhydrazones derived from alkyl trifluoromethyl ketone with (hetero)aromatic carbinols, please give some examples.
- 3) Regarding to the reaction mechanism, the authors stated that gem-difluorinated vinyl ether was formed from the HF elimination of oxonium ylide. It may be formed from the elimination of AgF from oxonium ylide. Please confirm this by experiments.

Reviewer #3 (Remarks to the Author):

This manuscript by Bi and coworkers details a C-F functionalization of trifluoroacyl groups to a wide range of difluoro allylic structures.

The manuscript showcases the applicability of this reaction to construct an impressive and diverse range of structural motives by applying either various heterocycles, allylic alcohols or propargylic alcohols as reaction partners. The functional group tolerance of this transformation is good and a large amount of products was included in the scope to demonstrate the utility. A mechanism was proposed, that was also probed by DFT computations. From the synthetic point I recommend publication of this manuscript in Nature Comms, however, I do have some reservations about the computational aspects of this work in the current state. Mostly, the method combination B3LYP/6-31G(d,p) really is outdated for relative reaction (free) energies. Firstly, there is no reason not to apply empirical dispersion correction (D3) as it comes for free and has been shown countless times to systematically improve energies with B3LYP and many other functionals, particularly for non-covalent interactions. This is of relevance to this work because the authors invoke non-covalent interactions to explain parts of the mechanism. Moreover, energies at the double-zeta level are less accurate than those obtained with a larger basis set, which is why single point calculations on the stationary points with a triple-zeta basis set are the commonly used standard procedure. Thus, I urge the authors to improve the relative (free) energies for the reaction profile. Possibly this will not change the overall conclusions from the calculations, but it will make the reaction barriers that the authors also quote in the text much more reliable.

Furthermore, I would appreciate the analysis of alternative reaction mechanisms in order to further back up the proposed mechanism. Alternatives that come to mind include:

- beta-F-elimination from Int2
- deprotonation of Int2 (the O-H⁺ should be highly acidic), possibly followed up by beta-F-elimination too
- reaction via an actual oxonium ylide, i.e. dissociation of [LAg] from Int2

Other more minor points:

- In my understanding, the intermediate after nucleophilic addition (e.g. bottom row, middle in Fig 1 or Int2 in Fig5) is not an ylide (defined as a charge separation at neighboring atoms) as long as Ag is bound to that carbon, but it is referred to as an ylide all throughout the manuscript
- Figure 1: I suggest clarifying the uncommon abbreviation Tf in the caption
- Figure 1, mechanism: the nucleophilic addition should be called just that and not "initiation", which is

a term commonly associated with chain reactions. Also, the electron-pushing arrow should not originate from Ag but from its bond to C

- Figure 4: please clarify the abbreviations DFHZ, TFHZ

- Figure 5: please add the structure of Int1

- Figure 5, the first part of the mechanism should be referred to as nucleophilic addition

- page 10, top: "these results suggest that the Ag catalyst plays a critical role [...] in the rearrangement process". Considering that the isolated intermediate 209 reacted to the product in 64% yield in the absence of Ag, the role cannot be so critical but rather supportive, please adjust the statement accordingly.

- page 10, middle/bottom: "Silver catalysis of a [3,3] rearrangement is, to our knowledge, without precedent". I don't know if this is really true but I don't think this is worth pointing out so much because I take it the reason is simply that Lewis-acid catalyzed rearrangements of this sort typically employ cheaper Lewis-acids than Ag-based ones.

REVIEWER COMMENTS

Reviewer #1 (Remarks to the Author):

Bi and co-workers reported a formal selective C-F bond allylation/allenylation of trifluoromethylhydrazones by Ag-catalysis to access a diverse range of α,α -difluoroalkylketones featuring heterocycles or alkene/allene moiety. A cascading process including Ag-mediated defluorinative oxygenation and fluorine-Claisen rearrangement was supposed as the success of the reaction, which is distinct from the well-known single electron manifold employed for analogous transformation of trifluoromethyl carbonyls. Significantly broad substrate scope was demonstrated with respect to both of the coupling partners. Also, the gram scale reaction and elaboration of the products highlight the synthetic potency of this reaction. Interestingly, control experiments and DFT calculation reveal the dual role of Ag catalyst to enable the formation of *gem*-difluoroalkene intermediate and facilitate the subsequent Claisen rearrangement. Overall, this review supports the publication of manuscript in Nat. Commun., after some revisions were addressed.

(1) Does fluorine atoms exert a positive effect on the [3,3]-rearrangement of *gem*-difluorovinyl ether? Could the reaction proceed under the same conditions when replacing fluoride with chloride or bromide?

(2) Some related work and reviews concerning rearrangements of fluorinated compounds should be cited, for example, *Tetrahedron Lett.* **1985**, 26, 2861; *J. Fluorine Chem.* **1999**, 100, 147; *J. Fluorine Chem.* **2004**, 125, 1593; *Tetrahedron* **2009**, 65, 9905; *Org. Biomol. Chem.* **2015**, 13, 7351, etc. Besides, a latest carbodefluorination reaction reported by the Feng group (*Angew. Chem. Int. Ed.* **2021**, 60, 20237), which also proceeded via a [3,3]-rearrangement, is suggested to be cited.

(3) In Figure 2, only indole, benzofurans, benzothiophenes, pyrrole, furan and thiophene derived carbinols were presented, what about substrates with greater degree of aromaticity, such as benzyl alcohol or naphthalene carbinol?

(4) The authors are suggested to comment on the driving force for the [3,3]-rearrangement which destroy the aromaticity of the substrate. Why the dearomatized products did not undergo rearrangement spontaneously to regain aromaticity?

(5) All presented alcohol substrates were primary and secondary alcohols, are tertiary alcohols viable substrates?

(6) Aside from alcohols, are other nucleophiles, such as amine and thiol suitable for this carbene insertion/rearrangement cascade?

(7) In the DFT part, the calculation result showed that the weak coordination of silver and oxygen could lower the energy barrier for the rate-determining rearrangement step. Could other stronger Lewis acids be applied instead of silver to further reduce the rearrangement energy barrier, resulting in even milder reaction conditions for the [3,3]-rearrangement of *gem*-difluorovinyl ether?

(8) While *N*-triflylhydrazone substrates derived from alkyl trifluoromethyl ketones reacted with β,γ -unsaturated alcohols readily, no such a case was demonstrated in Fig. 2 when reacted with heteroaryl carbinols, why?

(8) In cases **177-182**, why $\text{Rh}_2(\text{esp})_2$ exhibited higher efficiency than $\text{Tp}^{\text{Br}^3}\text{Ag}$? Does any alkene product derived from deprotonation of silver carbene was observed?

(9) In Feng's work (*ACIE* **2021**, 60, 20237), the [3,3]-rearrangement tend to yield CF_2 -bridged 1,5-diene product. However, in this work, transformation of **3** to **198**, *gem*-difluoroalkene was obtained as the

rearranged product, please provide an explanation for this.

(10) Last sentence of the abstract, "...in term of..." should be "... in terms of...".

Reviewer #2 (Remarks to the Author):

Very recently the synthesis of fluorinated organic compounds using the acyl-CF₃ compounds (CF₃C(O)R) as fluorine source through carbodefluorination strategy has received a great attention and big progress has been made. But I think this strategy is meaningless because acyl-CF₃ compounds are not easily prepared. That is: the introduction of fluorine atom into organic compounds was a challenging and hard, although the chemistry of the selective defluorination is interesting, the synthesis of new fluorinated compound through defluorination of fluorinated compounds was the waste of fluorine source. In this manuscript, Bi and co-workers described the carbodefluorination of acyl-CF₃ compounds via fluoroalkyl *N*-trifosylhydrazones. This reaction proceeded through the silver-catalyzed sequential carbene generation, oxonium ylide formation, C–F bond cleavage, and eventual C–C bond formation through Claisen rearrangement of the resultant difluorovinyl ethers. A broad range of

(hetero)aryl/alkyl fluoroalkyl ketones and β,γ-unsaturated alcohols were amenable to this reaction. This silver-catalyzed protocol provided single-step access to skeletally and functionally diverse α-mono- and α,α-difluoro-γ,δ-unsaturated ketones. The chemistry described in this manuscript is novel and interesting. Furthermore this manuscript was organized and written well. As I mentioned above, this method has less potential application in medicinal chemistry. But in light of the novelty of chemistry, I recommended for publication of this manuscript in Nature Communications when the following comments were addressed.

- 1) The title of this manuscript was too general. What kind of compound for carbodefluorination should be given.
- 2) How about the reaction of *N*-trifosylhydrazones derived from alkyl trifluoromethyl ketone with (hetero)aromatic carbinols, please give some examples.
- 3) Regarding to the reaction mechanism, the authors stated that *gem*-difluorinated vinyl ether was formed from the HF elimination of oxonium ylide. It may be formed from the elimination of AgF from oxonium ylide. Please confirm this by experiments.

Reviewer #3 (Remarks to the Author):

This manuscript by Bi and coworkers details a C-F functionalization of trifluoroacyl groups to a wide range of difluoro allylic structures.

The manuscript showcases the applicability of this reaction to construct an impressive and diverse range of structural motives by applying either various heterocycles, allylic alcohols or propargylic alcohols as reaction partners. The functional group tolerance of this transformation is good and a large amount of products was included in the scope to demonstrate the utility. A mechanism was proposed, that was also probed by DFT computations. From the synthetic point I recommend publication of this manuscript in Nature Comms, however, I do have some reservations about the computational aspects of this work in the current state. Mostly, the method combination B3LYP/6-31G(d,p) really is outdated for relative reaction (free)

energies. Firstly, there is no reason not to apply empirical dispersion correction (D3) as it comes for free and has been shown countless times to systematically improve energies with B3LYP and many other functionals, particularly for non-covalent interactions.

This is of relevance to this work because the authors invoke non-covalent interactions to explain parts of the mechanism. Moreover, energies at the double-zeta level are less accurate than those obtained with a larger basis set, which is why single point calculations on the stationary points with a triple-zeta basis set are the commonly used standard procedure. Thus, I urge the authors to improve the relative (free) energies for the reaction profile. Possibly this will not change the overall conclusions from the calculations, but it will make the reaction barriers that the authors also quote in the text much more reliable.

Furthermore, I would appreciate the analysis of alternative reaction mechanisms in order to further back up the proposed mechanism. Alternatives that come to mind include:

- beta-F-elimination from **Int2**
- deprotonation of **Int2** (the O-H+ should be highly acidic), possibly followed up by beta-F-elimination too
- reaction via an actual oxonium ylide, i.e. dissociation of [LAg] from **Int2**

Other more minor points:

- In my understanding, the intermediate after nucleophilic addition (e.g. bottom row, middle in Fig 1 or **Int2** in Fig5) is not an ylide (defined as a charge separation at neighboring atoms) as long as Ag is bound to that carbon, but it is referred to as an ylide all throughout the manuscript
- Figure 1: I suggest clarifying the uncommon abbreviation Tfs in the caption
- Figure 1, mechanism: the nucleophilic addition should be called just that and not "initiation", which is a term commonly associated with chain reactions. Also, the electron-pushing arrow should not originate from Ag but from its bond to C
- Figure 4: please clarify the abbreviations DFHZ, TFHZ
- Figure 5: please add the structure of **Int1**
- Figure 5, the first part of the mechanism should be referred to as nucleophilic addition
- page 10, top: "these results suggest that the Ag catalyst plays a critical role [...] in the rearrangement process". Considering that the isolated intermediate **209** reacted to the product in 64% yield in the absence of Ag, the role cannot be so critical but rather supportive, please adjust the statement accordingly.
- page 10, middle/bottom: "Silver catalysis of a [3,3] rearrangement is, to our knowledge, without precedent". I don't know if this is really true but I don't think this is worth pointing out so much because I take it the reason is simply that Lewis-acid catalyzed rearrangements of this sort typically employ cheaper Lewis-acids than Ag-based ones.

Point-by-point response to reviewer comments

Manuscript ID: NCOMMS-22-03965-T

Title: Carbodefluorination via a carbene-initiated rearrangement strategy

Author(s): Linxuan Li, Xinyu Zhang, Yongquan Ning, Xiaolong Zhang, Binbin Liu, Zhansong Zhang, Paramasivam Sivaguru, Giuseppe Zanoni, Shuang Li, Edward A. Anderson and Xihe Bi*

Dear Reviewers,

Thank you very much for your suggestions. We have revised this manuscript according to your comments. The corrections in detail were given in the revised manuscript and were highlighted in yellow color. The detailed revision was listed as follows:

Reviewer: 1

Recommendation: Accept

Comments:

Bi and co-workers reported a formal selective C-F bond allylation/allenylation of trifluoromethylhydrazones by Ag-catalysis to access a diverse range of α,α -difluoroalkylketones featuring heterocycles or alkene/allene moiety. A cascading process including Ag-mediated defluorinative oxygenation and fluorine-Claisen rearrangement was supposed as the success of the reaction, which is distinct from the well-known single electron manifold employed for analogous transformation of trifluoromethyl carbonyls. Significantly broad substrate scope was demonstrated with respect to both of the coupling partners. Also, the gram scale reaction and elaboration of the products highlight the synthetic potency of this reaction. Interestingly, control experiments and DFT calculation reveal the dual role of Ag catalyst to enable the formation of *gem*-difluoroalkene intermediate and facilitate the subsequent Claisen rearrangement. Overall, this review supports the publication of manuscript in *Nat. Commun.*, after some revisions were addressed.

Response: We thank this reviewer for taking the time and effort to review our manuscript and give affirmation to this work.

1. Does fluorine atoms exert a positive effect on the [3,3]-rearrangement of *gem*-difluorovinyl ether? Could the reaction proceed under the same conditions when replacing fluoride with chloride or bromide?

Response: Many thanks for this valuable comment. According to the reviewer's suggestion, we tested the [3,3] rearrangement of *gem*-dichloroalkenyl ethers at different temperatures. The *gem*-dichloroalkenyl ether was converted to the rearranged product **212** in 82% and 4% yield at 80 °C and 40 °C, respectively. In contrast, the fluorine-substituted alkenyl ethers afforded rearranged product **86** in almost quantitative and 70% yield at 80 °C and 40 °C, respectively. These results indicated that the fluorine atom has a positive effect on the [3,3] rearrangement of the *gem*-difluoroalkenyl ethers. We have added the experimental results and spectra to the supplementary information.

2. Some related work and reviews concerning rearrangements of fluorinated compounds should be cited, for example, *Tetrahedron Lett.* 1985, **26**, 2861; *J. Fluorine Chem.* 1999, **100**, 147; *J. Fluorine Chem.* 2004, **125**, 1593; *Tetrahedron* 2009, **65**, 9905; *Org. Biomol. Chem.* 2015, **13**, 7351, etc. Besides, a latest carbodefluorination reaction reported by the Feng group (*Angew. Chem. Int. Ed.* 2021, **60**, 20237), which also proceeded via a [3,3]-rearrangement, is suggested to be cited.

Response: Thank you for pointing it out. The related work and reviews have been cited as Ref.48-53 in the revised manuscript.

3. In Figure 2, only indole, benzofurans, benzothiophenes, pyrrole, furan and thiophene derived carbinols were presented, what about substrates with greater degree of aromaticity, such as benzyl alcohol or naphthalene carbinol?

Response: Benzyl alcohol reacted with *N*-trifosylhydrazone **1a** produced *gem*-difluoroalkenyl ether **213** in high yield, which can not undergo [3,3] rearrangement to form product **215**. These results have been reported in our recent work (*Chem. Eur. J.* **2022**, 28, e202200280).

4. The authors are suggested to comment on the driving force for the [3,3]-rearrangement which destroy the aromaticity of the substrate. Why the dearomatized products did not undergo rearrangement spontaneously to regain aromaticity?

Response: Many thanks for the valuable suggestion. According to this suggestion, we have added related comments to the text as shown below:

“The driving force to destroy the aromaticity of the substrate comes from the flow of electrons during the opening of the six-membered ring transition state during the [3,3]-rearrangement⁶¹.”

The dearomatized products are stable enough to be isolated under column chromatography conditions, but they are easy to restore aromaticity under acidic conditions as shown in Figure 4d. This phenomenon was also observed in a recent work of [2,3] rearrangement of indole-based onium ylides (Nair, V. N. et al. *J. Am. Chem. Soc.* **2021**, *143*, 9016–9025).

5. All presented alcohol substrates were primary and secondary alcohols, are tertiary alcohols viable substrates?

Response: Thank you very much for the valuable questions. When tertiary alcohols were used as substrates, we observed less than 5% of the product **217**, and 92% of the substrate was recovered. The possible reason is that the structure of the tertiary alcohol has a larger resistance. These results were summarized in supplementary information. The related comments were added to the text as shown below:

“However, we observed that tertiary alcohols are not suitable for this transformation, possibly because the structures of tertiary alcohols are more sterically hindered during electrophilic attack.”

6. Aside from alcohols, are other nucleophiles, such as amine and thiol suitable for this carbene insertion/rearrangement cascade?

Response: Many thanks for this comment. Other nucleophiles, such as allylamines and allylthiols, are also suitable for this carbene insertion/rearrangement cascade. The related investigations are ongoing in our laboratory and will be reported in another paper.

7. In the DFT part, the calculation result showed that the weak coordination of silver and oxygen could lower the energy barrier for the rate-determining rearrangement step. Could other stronger Lewis acids be applied instead of silver to further reduce the rearrangement energy barrier, resulting in even milder reaction conditions for the [3,3]-rearrangement of *gem*-difluorovinyl ether?

Response: The reviewer's assumption is right. We first used Lewis acids stronger than silver as the catalysts, such as $\text{BF}_3 \cdot \text{Et}_2\text{O}$, AlCl_3 , to catalyze [3,3] rearrangement of *gem*-difluoroalkenyl ethers under mild conditions (40 °C). The results show that both $\text{BF}_3 \cdot \text{Et}_2\text{O}$ and AlCl_3 are better catalyst than $\text{Tp}^{\text{Br}^3}\text{Ag}$, and the rearrangement product **86** is obtained in 72% and 75% yields. In addition, the rearrangement product was only obtained in 11% yield without catalyst. Therefore, the use of strong Lewis acids instead of silver can reduce the rearrangement energy barrier, allowing [3,3] rearrangements to be carried out under milder conditions. Afterwards, we used $\text{BF}_3 \cdot \text{Et}_2\text{O}$ and AlCl_3 to catalyze the carbodefluorination reaction. The results showed that compound **86** could not be obtained. This illustrates that the dual role of $\text{Tp}^{\text{Br}^3}\text{Ag}$ (metal carbene catalyst and Lewis acid catalyst) for the carbodefluorination reaction is necessary.

(i) Comparison of different catalysts at 40 °C

(ii) Different Lewis acids catalyze the carbodefluorination of **1a**

8. While *N*-triftosylhydrazone substrates derived from alkyl trifluoromethyl ketones reacted with β,γ -unsaturated alcohols readily, no such a case was demonstrated in Fig. 2 when reacted with heteroaryl carbinols, why?

Response: Many thanks for the reviewer's comments. We performed experiments using alkyl *N*-trifluoromethylhydrazones and 3-indolylcarbinols under Rh₂(esp)₂ catalysis and the result shows that the substrate is hard to be consumed and no target product is obtained.

9. In cases **177-182**, why Rh₂(esp)₂ exhibited higher efficiency than Tp^{Br3}Ag? Does any alkene product derived from deprotonation of silver carbene was observed?

Response: The reviewer is right. Silver alkyl carbenes are easy to undergo 1,2-hydrogen shifts to form alkenes. Because the high electrophilicity of silver carbene resulted in intramolecular 1,2-hydrogen shifts is more favorable than intermolecular reaction with allyl alcohols. Therefore, in cases **177-182**, Rh₂(esp)₂ exhibited higher efficiency than Tp^{Br3}Ag. In contrast, the rhodium carbene is more favorable for the electrophilic attack of allyl alcohols under Rh₂(esp)₂ catalysis.

10. In Feng's work (*ACIE* **2021**, *60*, 20237), the [3,3]-rearrangement tend to yield CF₂-bridged 1,5-diene product. However, in this work, transformation of **3** to **198**, *gem*-difluoroalkene was obtained as the rearranged product, please provide an explanation for this.

Response: Many thanks to the reviewer's comment. As shown in below, the transformation of **3** to **198** undergoes two processes, including Wittig reaction and [3,3] rearrangement. Compound **198** is more thermodynamically stable than compound **197**, thus resulting in the *gem*-difluoroolefin **198** as the final product after up to 60 °C. In Feng's work, the resulting CF₂-bridged 1,5-diene product formed by electron flow through a six-membered cyclic transition state is thermodynamically stable.

11. Last sentence of the abstract, "...in term of..." should be "... in terms of..."

Response: Thank you for pointing it out and we have corrected it as suggested.

Reviewer: 2

Recommendation: Minor Revision

Comments:

Very recently the synthesis of fluorinated organic compounds using the acyl-CF₃ compounds (CF₃C(O)R) as fluorine source through carbodefluorination strategy has received a great attention and big progress has been made. But I think this strategy is meaningless because acyl-CF₃ compounds are not easily prepared. That is: the introduction of fluorine atom into organic compounds was a challenging and hard, although the chemistry of the selective defluorination is interesting, the synthesis of new fluorinated compound through defluorination of fluorinated compounds was the waste of fluorine source. In this manuscript, Bi and co-workers described the carbodefluorination of acyl-CF₃ compounds via fluoroalkyl *N*-triflylhydrazones. This reaction proceeded through the silver-catalyzed sequential carbene generation, oxonium ylide formation, C-F bond cleavage, and eventual C-C bond formation through Claisen rearrangement of the resultant difluorovinyl ethers. A broad range of (hetero)aryl/alkyl fluoroalkyl ketones and β,γ-unsaturated alcohols were amenable to this reaction. This silver-catalyzed protocol provided single-step access to skeletally and functionally diverse α-mono- and α,α-difluoro-γ,δ-unsaturated ketones. The chemistry described in this manuscript is novel and interesting. Furthermore this manuscript was organized and written well. As I mentioned above, this method has less potential application in medicinal chemistry. But in light of the novelty of chemistry, I recommended for publication of this manuscript in Nature Communications when the following comments were addressed.

Response: Thank you very much for the helpful comments and valuable suggestions on our work.

1. The title of this manuscript was too general. What kind of compound for carbodefluorination should be given.

Response: According to the reviewer's suggestion, we have changed the title to "Carbodefluorination of fluoroalkyl ketones via a carbene-initiated rearrangement strategy".

2. How about the reaction of *N*-triflylhydrazones derived from alkyl trifluoromethyl ketone with (hetero)aromatic carbinols, please give some examples.

Response: Thanks for the valuable comment. According to the reviewer's suggestion, we have tested the reaction of alkyl *N*-trifluoromethylhydrazone **1b** and indole-3-carbinol under Rh₂(esp)₂ catalysis. However, no desired product **215** was observed and more than 95% of indole-3-carbinol was recovered. These results

have been summarized in supplementary information. The related comments were added to the text as shown below:

“It should be noted that *N*-trifosylhydrazones derived from alkyl trifluoromethyl ketone are not suitable substrates, as undergo competitive 1,2-hydrogen shifts to form alkenes.”

3. Regarding to the reaction mechanism, the authors stated that *gem*-difluorinated vinyl ether was formed from the HF elimination of oxonium ylide. It may be formed from the elimination of AgF from oxonium ylide. Please confirm this by experiments.

Response: Thank you very much for the helpful comments. To confirm this possibility, we increased the catalyst loading from 10 mol% to 30 mol% and detected the reaction residuum by X-ray powder diffraction (XRD), which suggested that no AgF powder was observed. In addition, AgF could not catalyze the reaction of *N*-trifosylhydrazone with indole-3-carbinol. Finally, the energy barrier for the elimination of AgF is 22.2 kcal/mol, which is much higher than that of the the elimination of HF. These results suggest that *gem*-difluorinated vinyl ether was formed from HF elimination rather than from the elimination of AgF from silver-coordinated oxonium ylide.

(i) AgF instead of $\text{Tp}^{\text{Br}^3}\text{Ag}$ for carbodefluorination reaction

(ii) X-ray powder diffraction analysis

(iii) AgF instead of $\text{Tp}^{\text{Br}^3}\text{Ag}$ for carbodefluorination reaction

Reviewer: 3

Recommendation: Minor Revision

Comments:

This manuscript by Bi and coworkers details a C-F functionalization of trifluoroacyl groups to a wide range of difluoro allylic structures.

The manuscript showcases the applicability of this reaction to construct an impressive and diverse range of structural motives by applying either various heterocycles, allylic alcohols or propargylic alcohols as reaction partners. The functional group tolerance of this transformation is good and a large amount of products was included in the scope to demonstrate the utility. A mechanism was proposed, that was also probed by DFT computations. From the synthetic point I recommend publication of this manuscript in Nature Comms, however, I do have some reservations about the computational aspects of this work in the current state. Mostly, the method combination B3LYP/6-31G(d,p) really is outdated for relative reaction (free) energies. Firstly, there is no reason not to apply empirical dispersion correction (D3) as it comes for free and has been shown countless times to systematically improve energies with B3LYP and many other functionals, particularly for non-covalent interactions.

This is of relevance to this work because the authors invoke non-covalent interactions to explain parts of the mechanism. Moreover, energies at the double-zeta level are less accurate than those obtained with a larger basis set, which is why single point calculations on the stationary points with a triple-zeta basis set are the commonly used standard procedure. Thus, I urge the authors to improve the relative (free) energies for the reaction profile. Possibly this will not change the overall conclusions from the calculations, but it will make the reaction barriers that the authors also quote in the text much more reliable.

Response: We highly appreciate this reviewer for taking the time and effort to review our manuscript, whose positive comments and insightful suggestions especially on the reaction mechanism and DFT calculations helped us greatly to improve the quality of the work. According to the suggestions, we have used B3LYP-D3 method with def2tzvp basis set to correct the free energy. We found that the relative free energy decreased, but the energy barrier trend was similar to our previous outcome. The new energy barrier diagram is as follows:

Furthermore, I would appreciate the analysis of alternative reaction mechanisms in order to further back up the proposed mechanism. Alternatives that come to mind include:

1. beta-F-elimination from **Int2**

Response: Many thanks. According to the reviewer's advice, we have tried to do β -F elimination attempts. Unfortunately, we did not find the correct transition state through the transition state search. We did not get the intermediate with β -F elimination.

2. deprotonation of **Int2** (the O-H⁺ should be highly acidic), possibly followed up by beta-F-elimination too?

Response: Many thanks. According to the reviewer's advice, we searched the deprotonated β -F elimination transition state. First, we want to eliminate F through the interaction between K and F, but when K is placed close to F, it is easy to form KF products. When it is placed relatively far away, F can't be eliminated and the correct transition state is not found. Then, we tried to eliminate F through the interaction between F and H that comes from $K_2CO_3H^+$, but failed to find the correct transition state.

In addition, we also made the transition state of HF elimination with the assistance of K_2CO_3 , and the energy barrier is 15 kcal/mol, which is higher than that without the assistance of K_2CO_3 (11.8 kcal/mol). The transition state structure is shown in the figure as below.

Finally, the transition state of the elimination of AgF is made, and the energy barrier is 22.2 kcal/mol, which is higher than that of the main pathway. The transition state structure is shown in the figure as below.

3. reaction via an actual oxonium ylide, i.e. dissociation of [LAg] from **Int2**

Response: According to the reviewer's advice, we calculated the route from **Int2** to free ylide. Since there is no ligand exchange, the energy barrier of direct dissociation of $\text{Tp}^{\text{Br}^3}\text{Ag}$ is large (46 kcal/mol), which is hard to occur.

4. In my understanding, the intermediate after nucleophilic addition (e.g. bottom row, middle in Fig 1 or **Int2** in Fig5) is not an ylide (defined as a charge separation at neighboring atoms) as long as Ag is bound to that carbon, but it is referred to as an ylide all throughout the manuscript

Response: Many thanks to you for pointing out this error. We consulted the relevant literature and similar intermediates of **Int2** should be referred to as "metal ylide" or "silver-coordinated oxonium ylide", and we have revised these in the main text. (*Science* **2019**, *366*, 990–994; *J. Am. Chem. Soc.* **2021**, *143*, 9016–9025.)

5. Figure 1: I suggest clarifying the uncommon abbreviation Tfs in the caption

Response: Many thanks. We have redrawn Fig. 1c and clarified the abbreviation Tfs. Please see Figure 1. in our revised manuscript.

6. Figure 1, mechanism: the nucleophilic addition should be called just that and not "initiation", which is a

term commonly associated with chain reactions. Also, the electron-pushing arrow should not originate from Ag but from its bond to C

Response: Many thanks. We have changed "initiation" in Figure 1 to "nucleophilic attack", and changed the direction of the arrow to start from the Ag-C bond.

7. Figure 4: please clarify the abbreviations DFHZ, TFHZ

Response: According to this suggestion, we have defined DFHZ-Tfs as difluoroacetaldehyde *N*-triftosylhydrazone and TFHZ-Tfs as trifluoroacetaldehyde *N*-tfsylhydrazone in Figure 4.

8. Figure 5: please add the structure of **Int1**

Response: According to this suggestion, we have added the structure of **Int1** in Figure 5.

9. Figure 5, the first part of the mechanism should be referred to as nucleophilic addition

Response: Many thanks. We have changed "Oxonium ylide formation" to "Nucleophilic attack" in Figure 5.

10. page 10, top: "these results suggest that the Ag catalyst plays a critical role [...] in the rearrangement process". Considering that the isolated intermediate 209 reacted to the product in 64% yield in the absence of Ag, the role cannot be so critical but rather supportive, please adjust the statement accordingly.

Response: Many thanks. We have changed "These results suggest that the Ag catalyst not only plays a critical role in formation of the difluoroalkene intermediate, but also in the rearrangement process." to "These results suggest that the Ag catalyst plays a critical role in the formation of difluoroalkene intermediate, and also facilitates the rearrangement process."

11. page 10, middle/bottom: "Silver catalysis of a [3,3] rearrangement is, to our knowledge, without precedent". I don't know if this is really true but I don't think this is worth pointing out so much because I take it the reason is simply that Lewis-acid catalyzed rearrangements of this sort typically employ cheaper Lewis-acids than Ag-based ones.

Response: Many thanks. We have deleted the sentence in our revised manuscript.

Finally, we would like to show our great respect to Referees. Their critical reviews and invaluable suggestions definitely have improved the quality of this manuscript.

Thank you once again for your consideration and efforts. We look forward to hearing from you.

Sincerely yours,

Xihe Bi

REVIEWERS' COMMENTS

Reviewer #1 (Remarks to the Author):

Given that the concerns raised by the reviewers have been properly addressed, the manuscript is recommended to be published as it is.

Reviewer #2 (Remarks to the Author):

As my comments were fully addressed, the revised manuscript was recommended for publication in Nature Communication.

Reviewer #3 (Remarks to the Author):

I thank the authors for taking the time to improve the DFT results. My questions and comments have all been addressed and I recommend publication of this manuscript in Nature Communications.